# How much do opinions regarding cultivated meat vary within the same country? The cases of São Paulo and Salvador, Brazil

**Gabriel Mendes**[1]*, **Jennifer Cristina Biscarra-Bellio**[1]*, **Marina Sucha Heidemann**[2], **Cesar Augusto Taconeli**[3], **Carla Forte Maiolino Molento**[1,2]

1 Cellular Animal Science Laboratory, Federal University of Paraná, Curitiba, Paraná, Brazil, 2 Animal Welfare Laboratory, Federal University of Paraná, Curitiba, Paraná, Brazil, 3 Department of Statistics, Federal University of Paraná, Curitiba, Paraná, Brazil

☯ These authors contributed equally to this work.
* gabriel_mds17@hotmail.com (GM); jenniferbiscarra@gmail.com (JCB-B)

## Abstract

The problems related to conventional meat production have been widely discussed globally and alternative proteins emerge as more sustainable and ethical options. Thus, understanding the intention to consume cultivated meat is key. This work aimed to study the intention to consume cultivated meat by residents of São Paulo and Salvador, Brazil, studying demographic differences. An online questionnaire comprising 17 multiple-choice and open-ended questions about opinions on conventional and cultivated meat was administered. The results were analyzed using quantitative methods, including binary logistic regression and ordinal regression models, as well as the qualitative Collective Subject Discourse methodology. With 809 participants, 419 (51.8%) from São Paulo and 390 (48.2%) from Salvador, 265 (32.8%, of which 170 (64.2%) from São Paulo and 95 (35.8%) from Salvador) respondents stated they would eat cultivated meat. Residents of São Paulo demonstrated higher familiarity with cultivated meat (187 (44.6%) had heard of it compared to 123 (31.5%) in Salvador). Such disparity in awareness seems coherent with differences in access to information and educational levels. Our results suggest that the acceptance of cultivated meat varies significantly across different regions of Brazil, likely related to the country's continental size, uneven economic and educational status and rich cultural diversity. We conclude that the acceptance of cultivated meat correlates with knowledge about it and that efforts to raise such knowledge require the consideration of cultural and socioeconomic aspects on a regional rather than national level, especially for geographically big and culturally diverse countries. Continued research is essential due to dynamics of acceptance and its entanglement with familiarity and knowledge regarding cultivated meat.

## Introduction

The projections by the United Nations, in 2022, suggest that the global population is likely to grow to around 8.5 billion in 2030, 9.7 billion in 2050 and 10.4 billion in 2100 [1]. The effect

**Data availability statement:** All S1 Table and S2 Table files are available in the FigShare database (accession: https://figshare.com/s/e296dea0c9ae16be6958).

**Funding:** This work was supported by the Forensic Sciences Project, CAPES, for data collection during the market and opinion research, and by the Fundação Araucária for the article processing charge. Gabriel Mendes was recipient of a scientific initiation scholarship from Fundação Araucária, Paraná, Brazil; Jennifer Cristina Biscarra Bellio was recipient of a CAPES doctorate scholarship; Marina Sucha Heidemann was recipient of a CAPES master's scholarship; Carla Forte Maiolino Molento was recipient of a productivity grant from CNPq.

**Competing interests:** We, the authors of the manuscript entitled 'Do opinions regarding cultivated meat vary within the same country? The case of Brazil', declare no conflict of interests, as also stated in the manuscript file.

of this overall human population growth on meat consumption is amplified by the progressive meatification of diets [2] in highly populated areas. Coherently, the demand for meat remains on the rise globally, leading to increased societal concerns about intensified animal production and, consequently, environmental and animal welfare issues [3]. The problems related to animal production have been called 'elephants in the room', as they are disproportionately discussed. Recently, eight elephants were reported in the room: (1) the absurd reality that food for more than 80 billion farm animals is produced in an effort to feed 8 billion humans, (2) forest destruction for pasture and crops to feed farm animals, (3) greenhouse gas emissions, (4) water use and waste management problems, (5) the unfair treatment of animals in intensive production systems, (6) the inefficiency of the current animal production systems as compared to new technologies, (7) public health risks of novel pandemics and antibiotic resistance, and (8) the predominant use of Earth's carbon atoms by human and, mostly, farm animal bodies [4].

In this sense, the problems involved in the current scenario of animal production indicate a need to adapt to more sustainable systems that also value aspects related to animal welfare [5], with concerns about such topics growing among consumers [6]. Coherently, concerns about the ethics and environmental impacts of traditional meat consumption have stimulated the development of alternatives [5], and protein source diversification emerges as a promising option, with the potential to generate environmental benefits [7] and higher standards regarding animal suffering [6]. Furthermore, alternative proteins address issues related to public health, food safety, and animal welfare [5,6,8–10]. However, there are challenges for their development. For instance, although cultivated meat uses significantly less land and water than conventional production, it may require higher energy inputs, depending on which specific chains are compared [11]. The climate effects of cultivated meat can also vary depending on the energy source used [8]. Biotechnological challenges for scaling up cultivated meat production in a cost-effective way remain [12], while the use of animal-free inputs presents its own set of research questions [13]. Yet another example is the need to understand consumer demands and intention to purchase cultivated meat products [14], as an evident essential part for the success of the new food production systems. Considering the remaining difficulties, the robust and multidimensional perceived benefits of cultivated meat and other alternative proteins boost increasing private and public investments for the resolution of the remaining challenges.

As a response to the need for science-based solutions, a new field of research, education, outreach activities and production systems conceived to develop and deliver alternative proteins, commonly denominated cellular agriculture, is currently under development [15]. In turn, alternative proteins are foods produced through the cultivation of animal cells *ex vivo* (i.e. out of the body), fermentation processes such as biomass and precision fermentation, or a combination of plant ingredients, with the goal of offering animal products or their analogues without the need for animal farming and slaughter [16]. Cultivated meat is a product of the culture of muscle tissue from a collection of cells from a live animal [17].

Some analogues to products of animal origin, such as those from plants, have been in human meals for decades. Whereas plant-based products are already commercialized, there are increasing levels of investment in cellular agriculture to improve knowledge regarding the existing technical, regulatory, and commercial challenges before all sorts of alternative proteins become widely available in the market [18], with price parity with their conventional counterparts. This price-parity seems particularly difficult to achieve unless fundamental shifts in subsidies currently available for conventional meat are enacted [19].

It is known that price is a major driving force for consumer buying decisions [20]. Would then price-parity between alternative proteins and conventional animal products suffice for a

major market shift towards alternative proteins? Maybe not. Meat consumers are reported to be resistant to alternative proteins due to a variety of reasons. An important reason seems to be their perception of risks in novel foods [18]. For instance, consumer preferences and risk perceptions influence the intention to try alternative proteins, and the greater the risk perceived the less likely consumers are to try them [21]. In addition, consumers who are skeptical when it comes to the decision of consuming cultivated meat report feelings of disgust, lack of naturalness, and negative sensory expectations [22–27]. Food neophobia, defined as an individual's reluctance to consume new foods [18], is a general reason for rejecting any food item on the basis of its novelty and unfamiliarity which also plays a role for rejecting alternative proteins.

An approach related to social sciences brings additional understanding to the potential acceptance of cultivated meat and other alternative proteins. Dietary behaviors are intrinsically linked to the social and economic expression of identities and preferences, playing a crucial role in nutritional status and health. Such links include the cultural context, which influences the place that different foods occupy within local culinary traditions and the perception of specific food technologies [28,29]. Within this complex multifactorial scenario composing food choices, it seems necessary to gain a more nuanced understanding of the psychological mechanisms that contribute to attitudes to and engagement with cultivated meat [30]. It is perceived that consumer acceptance will dictate its success [6,30], in addition to being a determinant of the new balance between conventional and cultivated meat production chains. In this sense, whether cellular agriculture will become more than just a niche market [31] depends critically on the percentage of consumers that buy its products.

It is logical, then, that the intention to purchase cultivated meat is a relevant research topic. In fact, the intention to consume cultivated meat has been the focus of scientific studies over the past decade, with the first publications appearing in 2015 [32]. The levels of acceptance of cultivated meat vary among different research results worldwide [22,33–35], as in Brazil [31,36–38]. However, treating consumption intention on a country basis seems an oversimplification that may hinder advancements in the field. This is especially relevant for countries with diverse cultural characteristics, which may be exacerbated by extensive territorial area and large human population sizes, as can be epitomized by the case of Brazil.

Brazil hosts around half the South American population and around half its geographical area. In addition, Brazil is the largest exporter of beef and the second largest producer of meat in the world [39], being arguably considered a country with a barbecue culture [36]. As agriculture and animal production constitute a major component of the country's Gross Domestic Product (GDP), the introduction of meat alternatives nationally, e.g. cultivated meat, seems essential to maintain its market share in the future, requiring careful planning to maximize the benefits and mitigate the disadvantages [40]. Thus, a more detailed regional understanding of Brazilians' perceptions of and intention to consume alternative proteins, with their underlying reasons, seems both relevant and a rich study case scenario. Brazilian regions differ significantly in geographic, socioeconomic, food culture, and media exposure aspects and it is likely that such differences significantly influence consumer acceptance of new foods. Indeed, in a previous study comparing two Brazilian cities within the same geographical region of Brazil, the South, significant differences in consumption intention were observed [31], raising interesting scientific questions regarding the magnitude of differences which may appear when the comparison is expanded to include cities from other regions in the country.

Salvador, the largest urban center in the Northeast region and the capital of the State of Bahia, with a population of 2,417,678 inhabitants [41], is currently ranked with the weakest economic and employment indicators among Brazilian State capitals [42]. In contrast, the city of São Paulo, capital of the Southeastern State of São Paulo, is the most populous city in Brazil,

with a population of 11,451,99 inhabitants [41], and stands out as the most economically developed center in the country [42].

This study explores consumption intentions and perceptions related to cultivated meat among residents of São Paulo and Salvador, two cities chosen for their large populations and striking regional contrasts, making them representative of their respective geographical areas. The research examines participants' knowledge and willingness to consume cultivated meat, their general perceptions of both conventional and cultivated meat, and their views on the relationship between these types of meat and key issues such as environmental impact, human health, and animal welfare. The primary aim is to understand Brazilians' intention to consume cultivated meat, with a particular focus on regional differences.

The paper begins by providing a demographic profile of the participants, followed by a detailed presentation of descriptive and qualitative data derived from their responses. These data are then summarized and analyzed using quantitative methods, including binary logistic regression and ordinal regression models, as well as the qualitative Collective Subject Discourse (CSD) [43] methodology. The analysis highlights comparisons between the two cities while considering demographic variables as potential explanatory factors. Finally, the discussion section addresses the study's findings, moving from general observations to more specific insights. It also outlines the study's limitations and suggests priorities for future research in the field.

## Material and methods

Residents of the cities of São Paulo and Salvador were invited to participate in an online survey. A consumer survey start-up (Opinion Box®) was hired to apply the questionnaire to individuals over 18 years-old, recruited randomly, from September 5th to 23rd, 2019. This work was approved by the Ethics Committee on Research with Humans, of Health Sciences Sector of the Federal University of Paraná, Brazil, and is registered under number 3.040.865/2018. To proceed with the questionnaire, participants were required to check a box indicating that they had read and agreed with the Informed Consent Form. This procedure ensured that they understood the study's objectives and nature before continuing. The explanation provided included details about the risks and benefits involved. They were also informed that they could withdraw from participation at any time, without the need to justify their decision and without any penalties, thus expressing their voluntary consent to participate in the study. The survey was elaborated based on a previous study of consumers in Southern Brazil [31] and composed of 15 multiple-choice and two open-ended questions. Questions 1 to 7 were related to sociodemographic data (gender, income, age, region, city, education, frequency of meat consumption), selected for their possible relationship with conventional meat consumption and the intention to consume cultivated meat, according to previous studies [13,22,29,31]. In addition to questions specifically about cultivated meat acceptance, we selected related issues that may provide insights into respondents' acceptance levels. The degree of familiarity with cultivated meat is likely a key factor in understanding consumer interest. Similarly, capturing respondents' perceptions of the positive and negative impacts of conventional meat on environmental, human health, and animal welfare issues may help interpreting their motivations for accepting or rejecting meat alternatives. Thereby, questions 8 to 17 (Table 1) explore participants' level of knowledge about conventional and cultivated meat, their intention to consume cultivated meat, and their perceptions of how the production of either one affects the environment, human health, and animal welfare. Questions 8 and 10 featured an image depicting conventional and cultivated meat chains. Questions from 12 to 17 presented the following response categories: very negative, negative, neutral, positive, and very positive.

**Table 1. Questions composing the online questionnaire administered from September 5 to 23, 2019, to residents of São Paulo and Salvador, Brazil.**

| Number | Question |
| --- | --- |
| 8 | What comes to your mind when you think of this conventional meat production system? Mention the first three words that come to your mind. |
| 9 | Have you heard about cultivated meat? |
| 10 | What comes to your mind when you think of this cultivated meat production system? Mention the first three words that come to your mind. |
| 11 | Would you eat cultivated meat? Justify |
| 12 | Regarding the environment, conventional meat is very negative/negative/neutral/positive/very positive |
| 13 | Regarding the environment, cultivated meat is very negative/negative/neutral/positive/very positive |
| 14 | Regarding human health, conventional meat is very negative/negative/neutral/positive/very positive |
| 15 | Regarding human health, cultivated meat is very negative/negative/neutral/positive/very positive |
| 16 | Regarding animal welfare, conventional meat is very negative/negative/neutral/positive/very positive |
| 17 | Regarding animal welfare, cultivated meat is very negative/negative/neutral/positive/very positive |

Sociodemographic data (questions 1 to 7) were compared with census results by the Brazilian Institute of Geography and Statistics (IBGE) [41], to verify whether the sample was representative of the population of the cities where the questionnaire was applied. Question 7 intended to identify respondents in terms of meat consumption, a factor likely related to their opinions regarding alternative proteins.

The answers to the open-ended questions were analyzed by two researchers, who initially synthesized the phrases individually into preliminary central ideas using the Collective Subject Discourse (CSD) method [43]. First, each researcher conducted their own data analysis, selecting the terms that, in their view, best represented the central themes. Subsequently, both researchers met to compare and discuss the selected terminology, evaluating each term to decide which best represented the respondents' ideas. During these discussions, a 90.0% agreement rate was achieved between the analyses. In cases of divergence, the reasons behind each term's proposal were thoroughly examined. The researchers then consulted an online Portuguese dictionary and, afterwards, an online English dictionary [44] to ensure translation accuracy and the semantic precision of the terms. Finally, these terms were incorporated into the word clouds, where the most frequent terms appear in proportionally larger sizes. The selection of words for the cloud was based on two main criteria: their technical meaning and relevance in the context of the research, as well as their general meaning according to the dictionary. The justification for answers to question 11 were also analyzed by the two different researchers, grouping them by similarity, using the same methodology.

The following factors were considered as explanatory variables: city, income, education, and frequency of meat consumption, as per categories in Table 2. Knowledge about cultivated meat, consumption intention, and the relationships between cultivated and conventional meat with the environment, human health and animal welfare were the subjects addressed in the questions. The statistical methods used, such as odds ratios and proportional odds regression models, are appropriate for analyzing the relationship between independent variables and the

**Table 2. Demographic results by response categories of the 809 respondents from an online questionnaire administered from September 5 to 23, 2019, to residents of São Paulo and Salvador, Brazil.**

| Variables | Categories | São Paulo 419 (51.8%) | Salvador 390 (48.2%) | Total 809 |
|---|---|---|---|---|
| Gender | Men | 189 (45.1%) | 156 (40.0%) | 345 (42.6%) |
| | Women | 230 (54.9%) | 234 (60.0%) | 464 (57.4%) |
| Age | 18–29 years old | 120 (28.6%) | 140 (35.9%) | 260 (32.1%) |
| | 30–49 years old | 219 (52.3%) | 205 (52.6%) | 424 (52.4%) |
| | 50 or older | 80 (19.1%) | 45 (11.5%) | 125 (15.5%) |
| Income | Up to R$1996.00 | 107 (25.5%) | 166 (42.6%) | 273 (33.7%) |
| | R$1997.00 to R$4990.00 | 188 (44.9%) | 156 (40.0%) | 344 (42.5%) |
| | More than R$4990.00 | 124 (29.6%) | 68 (17.4%) | 192 (23.8%) |
| Education | Elementary or high school | 151 (36.0%) | 190 (48.7%) | 341 (42.2%) |
| | Higher education | 218 (52.0%) | 176 (45.1%) | 394 (48.7%) |
| | Postgraduate degree | 50 (12.0%) | 24 (6.2%) | 74 (9.1%) |
| Frequency of meat consumption | Does not eat meat | 15 (3.6%) | 7 (1.8%) | 22 (2.7%) |
| | 1–3 days a week | 190 (45.3%) | 212 (54.4%) | 402 (49.7%) |
| | 4–7 days a week | 214 (51.1%) | 171 (43.8%) | 385 (47.6%) |

likelihood of categorical events, like the intention to consume cultivated meat. Odds ratios measure the effect of explanatory variables on event probabilities between two groups, while proportional odds regression models are useful for comparing outcomes with more than two ordered categories. These models were chosen for their ability to handle categorical data and proportional responses, common in consumption and attitude studies. For the binary responses (yes or no) we fitted binary logistic regression models, and the results are presented through the odds ratios of a positive response and corresponding confidence intervals (95%), whereas the ordinal response variables (very negative, negative, neutral, positive, very positive) were analyzed by fitting proportional odds regression models, and the results are presented by means of the odds ratios for a more favorable (positive) response. The ordinal scale variables were grouped before fitting the regression models, by combining the two less favorable options as a negative response, and the two more favorable as a positive response. In this case, each ordinal variable presents three categories: negative, neutral, and positive. All questions contained the "I do not know" answer option. Thus, an examination was also carried out on how demographic factors influenced knowledge, or lack thereof, in relation to each of the questions. We present the non-adjusted results, based on univariate regression models, as well as the adjusted results, provided by multiple regression models. All conclusions are based on a 5% significance level. All analyses were carried out using the R software [45] for statistical computation, version 4.3.1. The R library ordinal was used to fit the ordinal regression models.

## Results

The questionnaire was filled by a total of 809 respondents, 419 (51.8%) from São Paulo and 390 (48.2%) from Salvador. The demographic results are presented in Table 2. The income range varied between the two cities: in São Paulo the most frequently declared income was from R$1,997.00 to R$4,990.00 (44.9%), while in Salvador, it was up to R$1,996.00 (42.6%). Respondents from São Paulo most frequently claimed complete or incomplete higher

education (52.0%), and from Salvador, partial or complete elementary or high school (48.7%). Most respondents from São Paulo stated that they consumed meat 4 to 7 days a week (51.1%), while in Salvador it is 1 to 3 days a week (54.4%).

When respondents were asked if they had heard about cultivated meat, 310 out of 809 participants (38.3%) answered yes (Fig 1). In São Paulo, 187 respondents (44.6%) reported familiarity with cultivated meat, compared to 123 (31.5%) in Salvador. Table 3 presents the results of this question broken down by demographic variables.

Overall, 265 participants (32.8%) expressed interest in eating cultivated meat, 211 (26.0%) indicated they would not consume it, and 333 (41.2%) responded "I don't know" (Fig. 1). In São Paulo, 170 respondents (40.6%) said they would eat cultivated meat, compared to 95 respondents (24.4%) in Salvador. Table 4 presents the results of this question, categorized by demographic variables.

Among the 265 who showed interest in consuming cultivated meat, the main justifications were related to curiosity (32.0%), reporting verbs such as to try, to taste, to know; animal welfare (26.0%), with phrases such as to spare the lives of animals, to reduce slaughter, to avoid the killing of animals; human health (9.0%), with justifications such as it seems healthy, it would not be harmful to human health, it seems more nutritious.

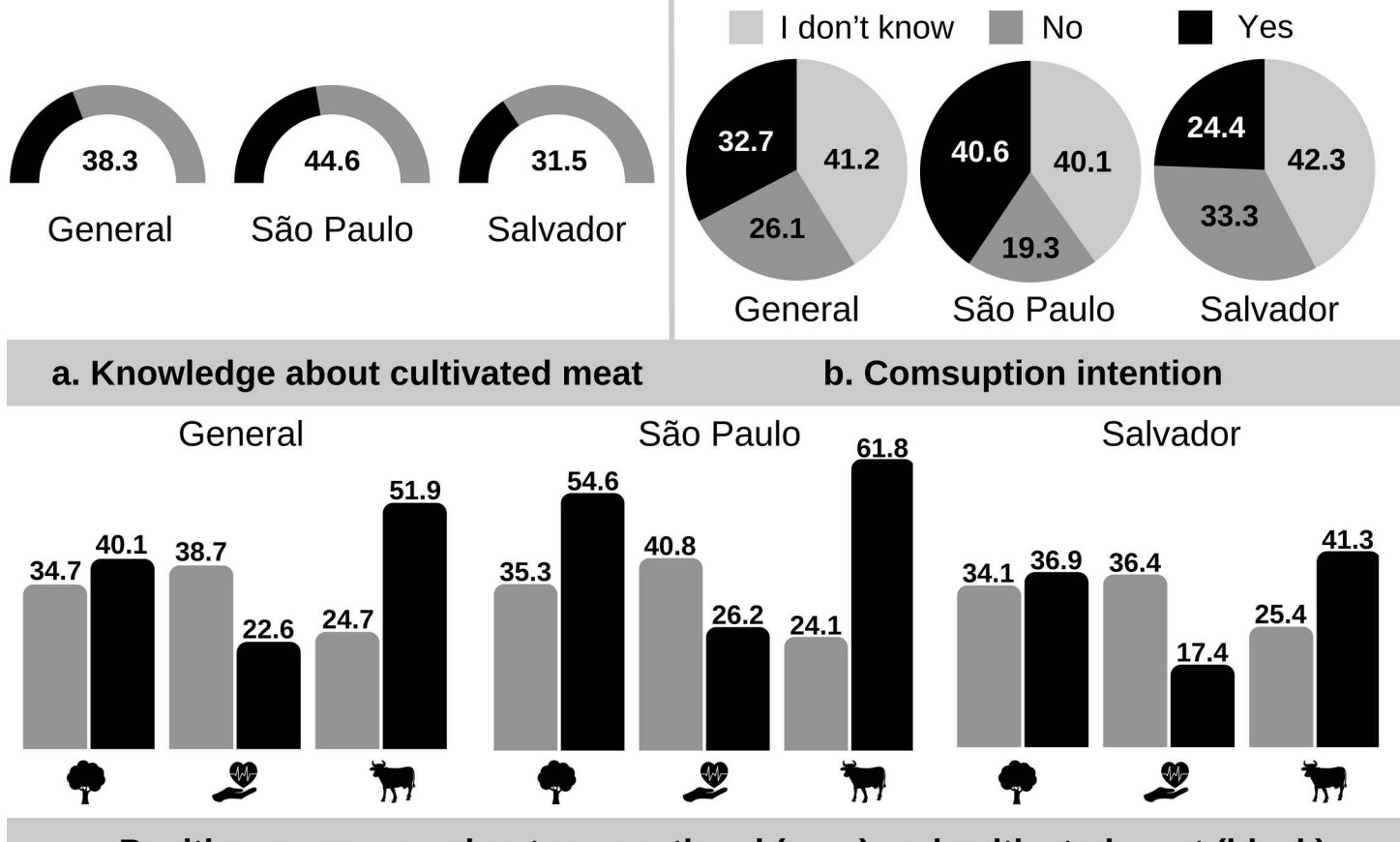

**Fig 1. General and by city percentages regarding knowledge about cultivated meat (a); the intention to consume cultivated meat (b); positive responses regarding the influence of conventional meat and cultivated meat on the environment, human health and animal welfare (c).**

**Table 3. Answers to the question "have you heard about cultivated meat?", given by 809 respondents from São Paulo and Salvador, Brazil, in an online questionnaire from September 5 to 23, 2019.**

| Variables | Categories | Have you heard about cultivated meat? | |
|---|---|---|---|
| | | Yes | No |
| City | São Paulo | 187 (44.6%) | 232 (55.4%) |
| | Salvador | 123 (31.5%) | 267 (68.5%) |
| Gender | Men | 146 (42.3%) | 199 (57.7%) |
| | Women | 164 (35.3%) | 300 (64.7%) |
| Age | 18–29 | 114 (43.8%) | 146 (56.2%) |
| | 30–49 | 151 (35.6%) | 273 (64.4%) |
| | 50 or older | 45 (36.0%) | 80 (64.0%) |
| Income | Up to R$1996.00 | 100 (36.6%) | 173 (63.4%) |
| | R$1997.00 to R$4990.00 | 120 (34.9%) | 224 (65.1%) |
| | More than R$4990.00 | 90 (46.9%) | 102 (53.1%) |
| Education | Elementary or high school | 113 (33.1%) | 228 (66.9%) |
| | Higher education | 161 (40.9%) | 233 (59.1%) |
| | Postgraduate degree | 36 (48.6%) | 38 (51.4%) |
| Frequency of meat consumption | Does not eat meat | 9 (40.9%) | 13 (59.1%) |
| | 1–3 days a week | 139 (34.6%) | 263 (65.4%) |
| | 4–7 days a week | 162 (42.1%) | 223 (57.9%) |

**Table 4. Answers to the question "would you eat cultivated meat", given by 809 respondents from São Paulo and Salvador, Brazil, in an online questionnaire from September 5 to 23, 2019.**

| Variables | Categories | Would you eat cultivated meat? | | |
|---|---|---|---|---|
| | | Yes | No | I do not know |
| City | São Paulo | 170 (40.6%) | 81 (19.3%) | 168 (40.1%) |
| | Salvador | 95 (24.4%) | 130 (33.3%) | 165 (42.3%) |
| Gender | Men | 128 (37.1%) | 79 (22.9%) | 138 (40.0%) |
| | Women | 137 (29.5%) | 132 (28.5%) | 195 (42.0%) |
| Age | 18–29 | 100 (38.5%) | 58 (22.3%) | 102 (39.2%) |
| | 30–49 | 137 (32.3%) | 110 (25.9%) | 177 (41.8%) |
| | 50 or older | 28 (22.4%) | 43 (34.4%) | 54 (43.2%) |
| Income | Up to R$1996.00 | 86 (31.5%) | 80 (29.3%) | 107 (39.2%) |
| | R$1997.00 to R$4990.00 | 104 (30.2%) | 92 (26.8%) | 148 (43.0%) |
| | More than R$4990.00 | 75 (39.1%) | 39 (20.3%) | 78 (40.6%) |
| Education | Elementary or high school | 109 (32.0%) | 101 (29.6%) | 131 (38.4%) |
| | Higher education | 131 (33.2%) | 93 (23.6%) | 170 (43.2%) |
| | Postgraduate degree | 25 (33.8%) | 17 (23.0%) | 32 (43.2%) |
| Frequency of meat consumption | Does not eat meat | 2 (9.1%) | 15 (68.2%) | 5 (22.7%) |
| | 1–3 days a week | 115 (28.6%) | 116 (28.9%) | 171 (42.5%) |
| | 4–7 days a week | 148 (38.4%) | 80 (20.8%) | 157 (40.8%) |

Regarding the justification of the 211 respondents who answered that they would not eat cultivated meat, the main ideas were grouped into i) neophobia (26.0%), with phrases such as I am afraid, dubious process, strange; ii) artificiality (21.0%), containing justifications such as not natural, too artificial, something modified; iii) lack of knowledge (15.0%), with ideas such as I do not know, lack of information about the product, I do not understand the process; and

iv) human health (15.0%), involving responses such as it is bad for our health, it is not healthy, it can cause disease.

The results of the most positive responses regarding the relationships between conventional meat and cultivated meat with the environment, human health, and animal welfare are represented in Fig 1.

Among the 310 respondents who had heard of cultivated meat, 156 (50.3%) said they would eat it, 64 (20.7%) said they would not, and 90 (29.0%) said they didn't know. Of the 499 respondents who were unfamiliar with cultivated meat, 109 (21.8%) said they would eat it, 147 (29.5%) said they would not, and 243 (48.7%) said they didn't know.

Among respondents who were already familiar with cultivated meat, 205 (66.1%) believe it benefits animal welfare, 108 (34.8%) consider it beneficial for human health, and 174 (56.1%) for the environment. In contrast, 215 (43.1%) among those who had never heard of cultivated meat, believe it benefits animal welfare, 75 (15.0%) see benefits for human health, and 199 (39.9%) for the environment (Table 5).

Regarding the odds ratios of providing a response other than "I don't know" response for each question, only those with statistical significance (95% CI, $p < 0.05$* or $p < 0.001$**) will be highlighted. Greater knowledge was observed among respondents from São Paulo compared to those from Salvador on the following questions: question 12 (OR = 1.95* in the bivariate analysis and OR = 1.79* in the multivariate analysis), question 13 (OR = 1.67* in the bivariate analysis and OR = 1.73* in the multivariate analysis), question 16 (OR = 3.16** in the bivariate analysis and OR = 3.10** in the multivariate analysis), and question 17 (OR = 1.69* in the bivariate analysis and OR = 1.77* in the multivariate analysis). Additional results are available in S1 Table.

The odds ratios of providing a more positive response for each question, comparing respondents from São Paulo to those from Salvador, are represented in Fig 2.

The respondents were asked to share the three words that came to their minds when thinking about the production systems of conventional and cultivated meat. Given the large volume of data, after analyzing each term, a word cloud was generated with the words that best represented the ideas expressed by the participants. For conventional meat, the words that best reflected the ideas mentioned by the participants were i) production (21.6%), with words like farm, production, and reproduction; the word "production" was chosen because it encompasses the raising of animals for food, including reproduction and the process of breeding species, which is central to the meat industry; ii) suffering (11.7%), with citations such as cruelty, torture, suffering; the word "suffering" was chosen because it encompasses the terms mentioned by respondents when expressing their opinions on the treatment of animals in the conventional production system; iii) logistics (11.6%), which includes words such as

Table 5. Responses to questions about the relationship between cultivated meat and the environment, human health, and animal welfare, based on respondents' familiarity of cultivated meat, collected from 809 participants in São Paulo and Salvador, Brazil, in an online questionnaire conducted from September 5 to 23, 2019.

| Have heard about | How cultivated meat relates to | Responses | | | |
|---|---|---|---|---|---|
| | | Beneficial | Neutral | Harmful | I do not know |
| Yes 310 (38.3%) | Animal Welfare | 205 (66.1%) | 54 (17.4%) | 33 (10.7%) | 18 (5.8%) |
| | Human Health | 108 (34.8%) | 77 (24.9%) | 62 (20.0%) | 63 (20.3%) |
| | Environment | 174 (56.1%) | 67 (21.6%) | 42 (13.6%) | 27 (8.7%) |
| No 499 (61.7%) | Animal Welfare | 215 (43.1%) | 114 (22.9%) | 84 (16.8%) | 86 (17.2%) |
| | Human Health | 75 (15.0%) | 144 (28.9%) | 114 (22.8%) | 166 (33.3%) |
| | Environment | 199 (39.9%) | 123 (24.7%) | 82 (16.4%) | 95 (19.0%) |

distribution, transport, logistics; the word "logistics" reflects the process related to the logistical aspects of the conventional meat production supply chain. For cultivated meat, the words that best reflected the ideas mentioned by the participants were i) innovation (12.1%), with words like technology, advance, innovative; the word "innovation" was highlighted due to its association with technological advances and the idea of something new and progressive mentioned by the respondents; ii) neophobia (10.8%), with words such as strange, fear, distrust; the word "neophobia" was chosen to represent the resistance and fear of new technologies expressed by the respondents; iii) artificial (8.2%), containing words such as false, unnatural, artificial; the word "artificial" was chosen as it is synonymous with the ideas of artificiality and unnaturalness expressed by the respondents (Fig 3).

The complete dataset, including all responses provided by respondents to each question in the questionnaire, is presented in S2 Table.

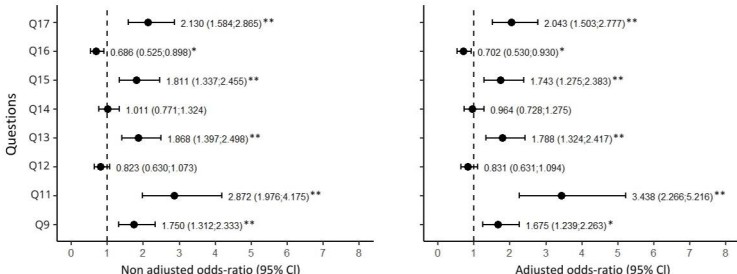

**Fig 2. Non-adjusted (bivariate) and adjusted (multiple) odds ratios for a more positive response among respondents from São Paulo compared to those from Salvador ($p < 0.05^*$, $p < 0.001^{**}$).**

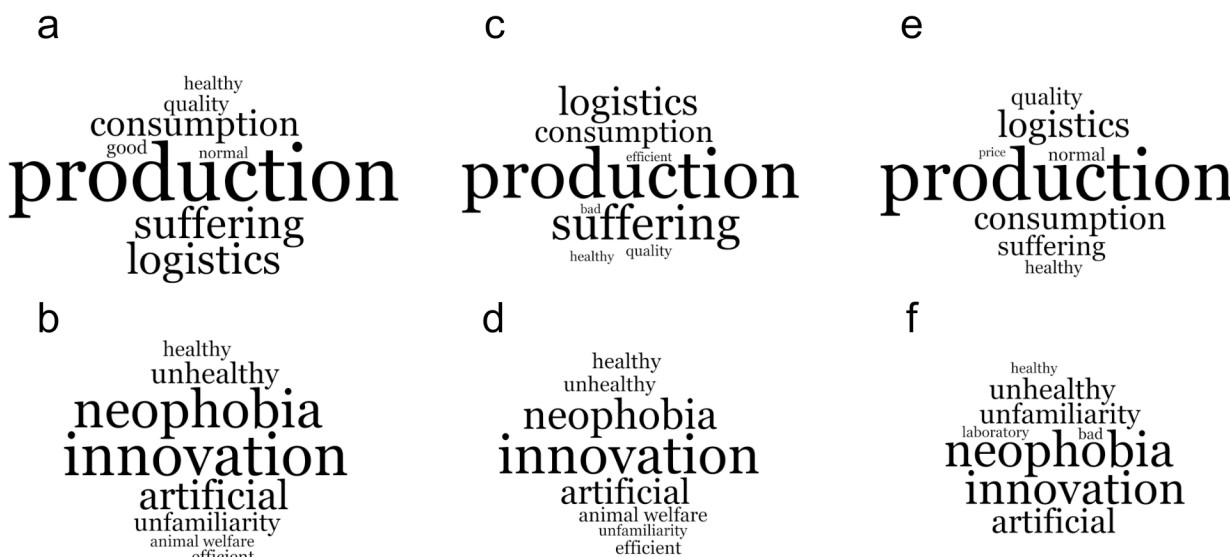

**Fig 3. Words representing the ideas expressed by 809 respondents from São Paulo and Salvador about conventional meat (a) and about cultivated meat (b); words representing the ideas expressed by 419 respondents from São Paulo about conventional meat (c) and about cultivated meat (d); words representing the ideas expressed by 309 respondents from Salvador about conventional meat (e) and about cultivated meat (f).**

## Discussion

Our results show that cultivated meat was not well known in Brazil in 2019, with respondents from Salvador demonstrating greater unfamiliarity compared to those from São Paulo. Concomitantly, conventional meat was more appealing to people from Salvador, those with lower income and with lower education. Salvador is the State capital city with the most fragile economic and employment rates in Brazil (where 16.0% of the population was unemployed in 2019), considering the comparison of 40 social indicators and the performance of municipal government in areas such as income and education, according to Sustainable City Institute [42]. Additionally, according to the general ranking of the Entrepreneurial Cities Index [42], which considers each of seven determinants—regulatory environment, infrastructure, market, access to capital, innovation, human capital, and culture—and their indicators, São Paulo leads as the most developed city in Brazil (with a score of 8.67), while Salvador appears in the 47th position (with a score of 6.05). Lack of familiarity with cultivated meat was cited as a barrier by participants [17]. Prior knowledge was also described as a driving factor for cultivated meat acceptance [33,46]. A study conducted in Germany with 713 participants showed that 38.0% of them already had prior knowledge of cultivated meat and demonstrated greater acceptance [33]. A study with 1,296 participants in the Netherlands indicated that 79.0% had never heard of cultivated meat. However, after explaining the technique and its possible advantages and disadvantages, 52.0% claimed to be willing to try it [47]. Additionally, a study conducted with 739 participants in the southern region of Brazil showed that 39.3% of them responded positively to the questions after watching an explanatory video about cellular agriculture [31]. Overall, the level of knowledge is embedded in a dynamic set of information and, as such, it is likely to improve rapidly as news about cultivated meat becomes increasingly frequent [31]. Salvador, with its higher unemployment rate and lower income levels, may have more limited access to information about new technologies, which in turn may influence the population's awareness and attitudes toward cultivated meat. It seems relevant to scrutinize the social inequalities in the regions studied to avoid biased interpretations that may suggest that people who reject cultivated meat do not like it, when in fact they may not have received adequate or any information about it.

This trend is reflected in the intention to consume cultivated meat, which was lower, with respondents from São Paulo showing greater interest. Studies conducted in Italy have shown cultivated meat consumption intentions of 54.0%, among 525 participants [34], while in Germany the rate was 57.0%, among 713 participants [33]. In the United States, the consumption intention was even higher, with 65.3%, among 673 participants [22]. Interestingly, in Brazil, intentions to consume cultivated meat were also higher in previous studies. Among residents of the Southern region of the country, consumption intentions of 63.6% [31] and 59.3% [36] were reported. In the Southeast region of Brazil, the consumption intention was even higher, reaching 80.9% among residents [37]. Therefore, the difference in acceptance among all studied regions of Brazil seems significant. Residents of São Paulo live in a city with a more diversified economy and greater access to technologies and innovations, which likely offers more opportunities for being constantly informed about new food alternatives, such as cultivated meat. Additionally, the higher purchasing power and greater environmental awareness in São Paulo may positively influence the acceptance of alternative products, like cultivated meat. In contrast, citizens of Salvador, facing economic challenges such as a higher unemployment rate and lower purchasing power, may encounter barriers to accessing information about these new products, which could result in a lower intention to consume. This is confirmed by our study, the first one to use the same methodology and simultaneous surveys to allow for a valid comparison between geographic regions.

In addition to regional differences in awareness, sensory perceptions also influence the acceptance of cultivated meat. Apparently, people are motivated to try cultivated meat mainly out of curiosity regarding its organoleptic properties, such as taste and texture. An important factor when considering whether the people would consume cultivated meat or not is whether it tastes the same as conventional meat [48]. Indeed, sensory quality is an important characteristic to increase consumer acceptability [20,49]. Nonetheless, flavor and texture can be adjusted to increase the acceptance of cultivated meat in the near and distant future [3]. The perception of the flavor of cultivated meat diverges from that of conventional meat. A study conducted with 214 participants in Canada, where 38% believed that cultivated meat would not have the same flavor as conventional meat, while 30% stated that it would have the same taste [48]. Although the flavor and texture of cultivated meat are unknown in Brazil and most people in the world, this does not prevent many people from wanting to eat it, with curiosity been a driving force according to our results.

Moving from sensory perceptions to broader motivations, animal welfare concerns also play a significant role in the growing interest in consuming cultivated meat. In fact, individuals from São Paulo exhibited a more positive view on this issue. Whether or not they were familiar with cultivated meat, respondents generally believed that it is more beneficial than harmful to animal welfare. The problems related to animal suffering in intensive production systems were exposed for the first time in 1964, with the publication of the book Animal Machines [50]. The welfare of animals raised in industrialized intensive farming and the naturalization of their slaughter are the main problems cited by Brazilians [14,31,51,52]. People perceive that the introduction of cultivated meat to the market may reduce, or at least prevent the worsening of, animal suffering caused by factory farming, as this alternative meat production system does not require the slaughter of animals [22,31]. Thus, cultivated meat emerges as an alternative for meat production without animal suffering and slaughter.

Alongside animal welfare, human health frequently drives the desire for cultivated meat. However, many respondents, particularly those unfamiliar with cultivated meat, believe conventional meat to be more health beneficial. As an important source of protein and other nutrients, meat is understood as a driver for global human health [53,54]. On the other hand, meat consumption is related to human health issues such as colon cancer [55], as well as cardiovascular diseases and type 2 diabetes [56]. Thus, the conflicting opinions registered by respondents seem to parallel scientific literature [23,46]. In terms of nutritional value, cultivated meat has the potential to be different to conventional meat, as its production allows for modifying the nutrient content, for instance, controlling the fat content and optimizing it with omega-3 fatty acids [35], in addition to having advantages such as the possibility of increasing its biological value by adding various vitamins, trace elements, amino acids, unsaturated fatty acids, and the fact that it cannot be a source of helminths [10,57] or many of the pathogens which may be present in conventional meat settings. Prior knowledge of the similarities between cultivated and conventional meat, differing mainly in their production methods, may help clarify the perception that cultivated meat is more harmful to human health than conventional meat. Therefore, some nutritional characteristics that impact human health, whether positive or negative, will be common to both types of meat. Further understanding the new possibilities for customizing cultivated meat according to specific health needs may bring higher perception of its potential benefits.

The environment was less frequently observed in our data as a reason for consuming cultivated meat. However, people believed that the relationship between cultivated meat and the environment is more positive than that of conventional meat, a perception that is especially evident among the inhabitants of São Paulo. Even among those who were not familiar with cultivated meat, many considered it more beneficial than harmful to the environment. Despite the importance of

protecting the environment, studies have shown that a minority of consumers are aware of the environmental impact of conventional meat [58]. Since animal production contributes to global environmental degradation, one approach is to decrease meat consumption [59], substituting it for meat analogues [60]. Cultivated meat is frequently presented as environmentally sounder. Compared to conventional meat production, cultivated meat is associated with approximately 7–45% less energy, 78–96% lower greenhouse gas emissions, 99% lower land use, and 82–96% lower water use, depending on the product compared [11,21,61]. Therefore, relatively high industrial energy demand has been estimated for cultivated meat [8]. Providing information about environmental benefits may increase the number of consumers in the market for cultivated meat [62], as even those who were not yet familiar with it perceived it positively in terms of its environmental impact. A barrier to this logic is that; however, for consumers to understand the environmental benefits of cultivated meat, more transparency on the environmental problems of conventional meat is required; however, such transparency is lacking and difficult to achieve [63]. However, all these data are estimates, as there is currently no large-scale global production of cultivated meat.

In terms of consumer demographics, respondents who consume meat more frequently, younger individuals, men, those with higher educational levels, and those with higher incomes have shown a more positive opinion towards cultivated meat. However, it seems difficult to predict which food protein items will be in higher demand in the next decades [14]. Cultivated meat is more attractive to young people in Brazil [38], which may be related to the fact that consumers who are younger, with higher level of education, wealthier, and politically progressive leaning are generally found to hold more positive views towards adopting novel food technologies [23,64]. At the same time, people who follow an omnivorous diet showed a strong intention to adopt alternative proteins, considering them as complementary to their regular diet [65]. Such positive opinions may be considered a starting point for the new food industry to design novel strategies of products in the perspective of the circular economy. These markers can contribute to the creation of an ecosystem that encompasses the social groups where people sympathetic to cultivated meat are found. Thus, marketing strategies can be targeted at this audience, as well as developing tactics to reach those who initially do not appear as enthusiasts of the new food systems.

While younger, wealthier, and more educated consumers tend to show more positive views towards cultivated meat, there are significant barriers to widespread acceptance, such as food neophobia. Neophobia, the perception of artificiality, and lack of familiarity with the product were reasons commonly mentioned for rejecting the consumption of cultivated meat. The level of food neophobia, a predictor of acceptance, is related to social conservatism towards the disruptive food products [66,67]. Furthermore, when consumers perceive food technology as unnatural, they are more likely to believe the technology is risky and are less prone to accept any potential benefits the technology may have [21,33,48]. Indeed, awareness of alternative protein sources is a psychological capability critical to shifting preferences away from meat. Having little or no awareness of alternative proteins is a significant barrier to consumption [28]. Democratizing access to information for unilingual populations requires efforts regarding the continuous offering of cultivated food terminology in local languages [16], especially given that the use of English terms may heighten perceptions of unfamiliarity among non-English speakers regarding cellular agriculture. The importance of increasing the use of local language is recognized, with translated terminology making access to new products and production systems more available to residents who do not speak foreign languages [23]. Decreasing conventional meat attachment and food neophobia, as well as increasing familiarity with and knowledge of the new food products, are important challenges to be overcome [68]. As cultivated meat has been perceived as having a high degree of unnaturalness, communications and marketing of cultivated products should strive to portray cultivated meat in a more natural and favorable light [69]. Educating society has always proved to be effective in overcoming psychological hindrance; therefore, communication channels need to be better exploited [67].

Consumers should be encouraged to become more familiar with the product, so they can reflect upon it with knowledge. One method of doing this is to use terminology and product labeling of a less technical nature [70]. As cultivated meat becomes more prevalent in the daily lives of consumers, the perception of fear and artificiality regarding the product may decrease.

As consumers become more familiar with and educated about cultivated meat, the transition from conventional to alternative protein sources may become smoother. This shift, however, is not without challenges. While conventional meat production is well established, cultivated meat represents a disruptive innovation not yet in the market. As new technologies for alternative proteins, like cultivated meat, are expected to emerge soon [40], replacing conventional meat may become easier, less taboo and even a social differentiation marker [71]. The individual transition may be in the form of a partial replacement of conventional meat, as seen in flexitarian diets [72]. However, the dominant conventional meat-centered culture tends to remain [73]. Our findings highlight the need for strategies to improve knowledge about cultivated meat, requiring tailored planning based on regional characteristics, even within the same country.

Some potential biases and limitations may have influenced the results of this work. The sampling, while extensive in two large Brazilian cities, may not fully represent the general population. The online questionnaire may have attracted more participants with internet access and familiarity with or at least curiosity about cultivated meat. In addition, the reliance on limited respondent stratification may relate to subgroups of the population not being represented in our survey, which requires caution in making inferences for the specific cities studied and even more so for the country in general. The limited number of questions in the survey, especially open-ended questions, is also a limitation, which combined with the need for extracting key words may result in an oversimplification of respondent views. The self-reported data may exhibit social desirability bias, where respondents provide answers that they consider socially acceptable, susceptible to personal interpretations and selective memory [74,75]. Additionally, the lack of prior knowledge about cultivated meat intrinsically limited the strength of rationales supporting the responses regarding the perceptions of its potential impacts regarding the environment, human health, and animal welfare. An additional intrinsic issue relates to the dynamic character of the acceptance of cultivated meat, even more marked as our results show the significance of knowledge and familiarity, both of which tend to increase with time. This means that the picture our data set shows is best understood as historical than as descriptive of today's scenario. These limitations underscore the complexities of gathering data and interpreting public opinion on cultivated meat, also exposing the need for continued research.

## Conclusion

The acceptance of cultivated meat varies significantly across Brazil, likely due to the country's continental size, economic disparity, and rich cultural diversity. This variation suggests that the dissemination of knowledge about cultivated meat may encounter barriers related to regional public understanding, accessibility, and preferences. Our results have direct implications, as they make such challenges evident, and expose the need to considerate regional identities when promoting cultivated meat, as resistance may stem from unfamiliarity or concerns about changes to food traditions. It remains to be studied whether variations emerge in countries with different geographical, demographic and cultural profiles.

Our findings highlight the importance of regional considerations not only for academic research but also for practical applications in public policy, industry practices, and consumer education initiatives. The efficiency of public policies will benefit from targeted educational initiatives that address the specific concerns of different regions, while industry practitioners might adjust their outreach strategies to be more culturally sensitive. In addition, collaborating with local institutions is likely to foster a more productive approach to introducing cultivated meat.

Future studies are warranted to explore the influence of different demographic factors on acceptance, assess the efficacy of targeted educational strategies in enlightening perceptions, and track opinion shifts over time as knowledge and availability of cultivated meat increase. Such research will be invaluable in fostering public understanding, addressing misconceptions, supporting public policies and facilitating informed decision-making regarding cultivated meat and other alternative proteins.

## Supporting information

**S1 Table**. **Dataset of odds ratios of responses for each question, comparing the variables gender, age, income, education, and meat consumption among respondents from São Paulo and Salvador, from September 5th to 23rd, 2019.**
(DOCX)

**S2 Table**. **Full dataset with the participation of respondents from São Paulo and Salvador, from September 5th to 23rd, 2019.**
(XLSX)

## Acknowledgments

The authors would like to thank all the anonymous participants who gave their time to complete the questionnaire for this study. We also thank two reviewers who significantly contributed to the refinement of our paper.

## Author contributions

**Conceptualization:** Marina Sucha Heidemann, Carla Forte Maiolino Molento.

**Data curation:** Gabriel Mendes, Jennifer Cristina Biscarra-Bellio, Marina Sucha Heidemann, Carla Forte Maiolino Molento.

**Formal analysis:** Gabriel Mendes, Jennifer Cristina Biscarra-Bellio, Cesar Augusto Taconeli, Carla Forte Maiolino Molento.

**Funding acquisition:** Carla Forte Maiolino Molento.

**Investigation:** Carla Forte Maiolino Molento.

**Methodology:** Carla Forte Maiolino Molento.

**Project administration:** Carla Forte Maiolino Molento.

**Resources:** Carla Forte Maiolino Molento.

**Supervision:** Carla Forte Maiolino Molento.

**Writing – original draft:** Gabriel Mendes, Jennifer Cristina Biscarra-Bellio.

**Writing – review & editing:** Marina Sucha Heidemann, Cesar Augusto Taconeli, Carla Forte Maiolino Molento.

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
