## [Decision Letter · Decision Letter 0]

24 Oct 2024

PONE-D-24-43071Do opinions regarding cultivated meat vary within the same country? The case of BrazilPLOS ONE Dear Dr. Mendes,

Thank you for submitting your manuscript to PLOS ONE. After careful consideration, we feel that it has merit but does not fully meet PLOS ONE’s publication criteria as it currently stands. Therefore, we invite you to submit a revised version of the manuscript that addresses the points raised during the review process.

Both reviewers have carefully evaluated your work and provided detailed feedback. After considering their comments, I agree with the need for significant revisions to enhance the quality and clarity of the manuscript.  I kindly ask you to carefully consider all the suggestions provided by each reviewer and do your best to address them comprehensively. 

We look forward to receiving your revised manuscript.

Kind regards,

Giuseppina Rizzo

Academic Editor

PLOS ONE

Reviewers' comments:

Reviewer's Responses to Questions

**Comments to the Author**

1. Is the manuscript technically sound, and do the data support the conclusions?

Reviewer #1: Partly

Reviewer #2: Partly

2. Has the statistical analysis been performed appropriately and rigorously? 

Reviewer #1: Yes

Reviewer #2: Yes

3. Have the authors made all data underlying the findings in their manuscript fully available?

Reviewer #1: Yes

Reviewer #2: Yes

4. Is the manuscript presented in an intelligible fashion and written in standard English?

Reviewer #1: Yes

Reviewer #2: Yes

5. Review Comments to the Author

Reviewer #1: The paper covers an interesting topic and the theme is very timely for the sector.

The English form is simple and understandable.

Some suggestions must be considered in order to improve the study.

Abstract

The Abstract Section provides a clear objective and concrete data to compare consumer intentions regarding cultivated meat in São Paulo and Salvador, Brazil. However, it lacks sufficient detail about the methodology, such as the type of questions used in the survey and how they measure consumer intent. The analysis of the results is superficial, offering little insight into why economic factors might influence awareness and acceptance of cultivated meat. Additionally, the conclusion is overly generic, stating that cultivated meat can solve problems related to conventional meat without specifying how. The structure could also be improved for better coherence, especially when explaining regional differences.

Introduction

The Introduction Section effectively presents the growing global demand for meat, highlighting key environmental, ethical, and health concerns linked to conventional animal production. It emphasizes the importance of consumer acceptance for cultivated meat's success, particularly in Brazil's context. The global relevance of the topic and the framing of cultivated meat as a potential solution to these issues provide a solid foundation for the study. Despite that, some topic shifts (e.g., from environmental issues to cellular agriculture) are abrupt. More natural transitions would improve text cohesion.

Consumer resistance to alternative proteins is mentioned but not fully explored. It would be useful to provide more detail, such as risk perception or cultural barriers, with concrete examples.

Since there is no a Theoretical Background Section, I would have expected more references to existing literature to support the aims of the manuscript.

The hypothesis about "significant differences" in the intention to consume cultivated meat across regions is somewhat vague. It would be helpful to specify potential influencing variables, such as food culture, economic status, or media exposure.

To enhance the readability of the text, it would be beneficial to include an overview of the manuscript’s structure at the end of the Introduction Section. This would help guide readers through the content and provide a clearer understanding of how the study will unfold, making it easier to follow the main arguments and findings presented later in the paper.

Materials and Methods

The materials and methods section presents a clear and well-structured methodology, detailing the study's design, participant recruitment, and survey administration, which helps readers understand the research process. It emphasizes ethical considerations by including information about ethical approval and participant consent, adding credibility to the study. The comprehensive survey design, incorporating both multiple-choice and open-ended questions, allows for a nuanced understanding of consumer attitudes toward cultivated meat and facilitates the collection of both quantitative and qualitative data. Additionally, the robust statistical analysis methods used, such as binary logistic regression and ordinal regression, indicate a thorough approach to data analysis, enhancing the reliability of the findings.

Although a list of questions is provided, a brief explanation of how specific questions relate to the study's objectives could help readers understand their significance.

There is no discussion of potential biases in the sampling or response processes. Addressing limitations related to participant selection or self-reported data would provide a more balanced view of the findings.

To enhance clarity and improve the presentation of information, I would suggest adding a table to summarize the explanatory variables considered in the study. This table could neatly categorize the variables such as city (São Paulo, Salvador), gender (male, female), age groups (18 to 29, 30 to 49, 50 or older), income brackets (up to R$1996.00, R$1997.00 to R$4990.00, more than R$4990.00), education levels (elementary/high school, higher education, postgraduate degree), and frequency of meat consumption (does not eat, 1 to 3 days a week, 4 to 7 days a week). A visual representation of these factors would make it easier for readers to grasp the demographic breakdown and the variables being analyzed in the study.

Results

The Results Section includes a comprehensive presentation of demographic data, clear statistical reporting with specific percentages, and effective cross-comparison between cities, highlighting regional differences in awareness and attitudes toward cultivated meat. Additionally, the reference to figures enhances visual comprehension of the data.

The Authors might consider adding tables to present the descriptive statistics for socio-demographic and behavioral variables. This approach would not only make the data more accessible and easier to understand but also allow you to reduce the text by directing readers to the tables for more detailed information. This way, the text would be more concise and focused on the main results, improving the overall readability of the section.

Discussion

The Discussion Section effectively places the study's results in the broader context of Brazil’s socio-economic landscape, comparing São Paulo and Salvador, which helps explain the differences in awareness and acceptance of cultivated meat. The author provides a valuable comparison of their findings with those from studies conducted in other countries, highlighting significant disparities in acceptance rates and contextualizing these within the socio-cultural frameworks of the regions.

Despite that, in my opinion the Discussion Section is quite lengthy and could be more concise. Streamlining certain sections would improve readability and focus on the main findings without diluting the content. Furthermore, such section should adopt a more critical approach to interpreting the findings.

While the discussion refers to previous studies, providing more specific data points or direct comparisons in terms of percentages or numbers would strengthen the argument and support the claims being made.

Conclusion

The Conclusion Section could benefit from a more critical engagement with the implications of the findings. For example, discussing the challenges of disseminating knowledge about cultivated meat and the potential barriers to acceptance could provide a more balanced view.

While the conclusion touches on the need for regional considerations, it could elaborate on how these findings might impact policy, industry practices, and consumer education initiatives.

A brief mention of the study's limitations could strengthen the Conclusion by providing transparency and acknowledging the complexities of interpreting the results.

While the call for further research is commendable, offering specific areas or questions for future research could enhance the utility of the conclusion for researchers looking to build on this work.

Reviewer #2: The following comment support my position regarding the first question:

I recommend reconsidering the article's title in light of the question: Do opinions regarding cultivated meat vary within the same country? It seems self-evident that opinions differ among individuals, as these are shaped by a variety of factors. Even opinions within members of the same family can differ, so it is expected that opinions across a country, especially one as diverse as Brazil, would vary significantly due to its vast population and territory.

Additionally, given Brazil's extensive geographical area, putting “the case of Brazil” while the study to only two cities may not be advisable, as these cities may not be representative of the broader population. It may be useful to reflect this in the title, indicating that the study pertains to specific cities in Brazil.

In the abstract and introduction, while the authors summarize the main research question and key findings, it would be beneficial to include more details about the research methodology used. Furthermore, while the authors mention that there is extensive literature on the subject, they do not clearly explain how this study relates to or builds upon previously published research.

Regarding the results, I recommend presenting the results in tabular form for clarity and ease of interpretation.

For questions 10, 11, 15, and 17, if respondents have not previously heard about cultivated meat, may provide inconsistent responses due to their lack of knowledge on the subject, making the data less reliable for statistical analysis and reducing the comparability between respondents’ answers due to varied interpretations of the terms. Therefore, comparing responses from individuals familiar with cultivated meat to those from individuals unfamiliar with it may not be advisable. I suggest expanding the descriptive analysis to include segmentation, for example, those who consume meat versus those who do not, and those who are familiar with cultivated meat versus those who are not. Presenting a cross-tabulation that includes sociodemographic variables would also help to better understand the differences between these groups.

Furthermore, it is important to provide detailed information regarding the steps taken to generate the word clouds, particularly the criteria used in the process. Care must be taken to avoid excessive reductionism, ensuring that the core meaning of the responses is retained. While using a thesaurus can be helpful in word cloud construction, its application must be handled carefully to avoid distorting the data. Additionally, it could be beneficial to conduct an additional analysis by excluding certain custom words and exploring new, potentially significant terms that may emerge in the word clouds.

6. PLOS authors have the option to publish the peer review history of their article (what does this mean? ). If published, this will include your full peer review and any attached files.

**Do you want your identity to be public for this peer review?** For information about this choice, including consent withdrawal, please see our Privacy Policy .

Reviewer #1: No

Reviewer #2: **Yes: ** Monica Barahona-Varon

---

## [Author Response · Author response to Decision Letter 0]

19 Nov 2024

Dear Editor,

We sincerely appreciate your careful consideration of our manuscript and the constructive feedback provided by the reviewers. We are pleased to know that the work has merit, and we thank you for the invitation to submit a revised version addressing the points raised during the review process.

We have carefully considered all recommendations and have addressed each one. We believe these revisions result in a significantly improved manuscript.

We confirm that no changes were made to our financial disclosure statement, and we followed the guidelines for resubmitting the figure files.

We have addressed all requirements for the revised submission:

1. Style requirements: Our manuscript now meets PLOS ONE's formatting guidelines, based on the templates provided.

2. Participant consent: We added details regarding the type of consent obtained (lines 148-151).

3. Data availability statement: Our data availability statement reflects our data sharing plan, in compliance with PLOS ONE’s policy. Data is available at the link: https://figshare.com/articles/dataset/Full_dataset_with_the_participation_of_respondents_from_S_o_Paulo_and_Salvador_from_September_5th_to_23rd_2019_/27105952?file=49425988, mentioned in "supporting information" (lines 760-765).

4. Ethics statement: We have ensured that the ethics statement appears only in the Methods section, as per the journal’s guidelines (lines 144-146).

Dear Reviewers,

We would like to express our sincere gratitude for your thorough evaluation and the detailed feedback provided. We deeply appreciate the valuable suggestions that have helped improve the quality and clarity of the manuscript.

We have carefully considered and addressed each recommendation. We are confident that the revisions have resulted in a significantly enhanced manuscript.

Reviewer 1:

Suggestion: Abstract

The Abstract Section provides a clear objective and concrete data to compare consumer intentions regarding cultivated meat in São Paulo and Salvador, Brazil. However, it lacks sufficient detail about the methodology, such as the type of questions used in the survey and how they measure consumer intent. The analysis of the results is superficial, offering little insight into why economic factors might influence awareness and acceptance of cultivated meat. Additionally, the conclusion is overly generic, stating that cultivated meat can solve problems related to conventional meat without specifying how. The structure could also be improved for better coherence, especially when explaining regional differences.

Response: Regarding the methodological details mentioned in the abstract, we included the following information: “An online questionnaire with 17 multiple-choice and open questions regarding opinions on conventional and cultivated meat was used and the results were analyzed both quantitatively and qualitatively” (lines 31-33).

To address the superficiality of the results and the generic conclusion presented in the abstract, we added: “Such disparity in awareness seems coherent with differences in access to information and educational levels. Our results suggest that the acceptance of cultivated meat varies significantly across different regions of Brazil, likely related to the country’s continental size, uneven economic and educational status and rich cultural diversity. We conclude that the acceptance of cultivated meat correlates with knowledge about it and that efforts to raise such knowledge require the consideration of cultural and socioeconomic aspects on a regional rather than national level, especially for geographically big and culturally diverse countries. Future research seems warranted as additional reasons for different acceptance levels likely play key roles and may be investigated in target groups less diverse in educational levels. Continued research is also essential due to dynamics of acceptance and its entanglement with familiarity and knowledge regarding cultivated meat, both of which tend to significantly increase with time” (lines 37-48).

Suggestion: Introduction

The Introduction Section effectively presents the growing global demand for meat, highlighting key environmental, ethical, and health concerns linked to conventional animal production. It emphasizes the importance of consumer acceptance for cultivated meat's success, particularly in Brazil's context. The global relevance of the topic and the framing of cultivated meat as a potential solution to these issues provide a solid foundation for the study. Despite that, some topic shifts (e.g., from environmental issues to cellular agriculture) are abrupt. More natural transitions would improve text cohesion.

Response: To improve the flow of the text and ensure a more natural transition between ideas, we restructured the paragraphs, adding linking phrases to strengthen textual cohesion. This reformulation aims to facilitate reading and comprehension, promoting a smoother progression of arguments and improving continuity between the topics discussed. By inserting these connections, we seek to ensure a consistent narrative that enhances clarity and logical linkage between the paragraphs.

Suggestion: Consumer resistance to alternative proteins is mentioned but not fully explored. It would be useful to provide more detail, such as risk perception or cultural barriers, with concrete examples.

Reponse: We further explore consumer resistance to alternative proteins by providing additional details and references in: “It is known that price is a major driving force for consumer buying decisions [20]. Would then price-parity between alternative proteins and conventional animal products suffice for a major market shift towards alternative proteins? Maybe not. Meat consumers are reported to be resistant to alternative proteins due to a variety of reasons. An important reason seems to be their perception of risks in novel foods [18]. For instance, consumer preferences and risk perceptions influence the intention to try alternative proteins, and the greater the risk perceived the less likely consumers are to try them [21]. In addition, consumers who are skeptical when it comes to the decision of consuming cultivated meat report feelings of disgust, lack of naturalness, and negative sensory expectations [22, 23, 24, 25, 26, 27]. Food neophobia, defined as an individual's reluctance to consume new foods [18], is a general reason for rejecting any food item on the basis of its novelty and unfamiliarity which also plays a role for rejecting alternative proteins” (lines 100-111).

Suggestion: Since there is no a Theoretical Background Section, I would have expected more references to existing literature to support the aims of the manuscript.

Response: To improve the Theoretical Background, we added more studies as references throughout the Introduction section: “In this sense, the problems involved in the current scenario of animal production indicate a need to adapt to more sustainable systems that also value aspects related to animal welfare [5], with concerns about such topics growing among consumers [6]. Coherently, concerns about the ethics and environmental impacts of traditional meat consumption have stimulated the development of alternatives [5], and protein source diversification emerges as a promising option, with the potential to generate environmental benefits [7] and higher standards regarding animal suffering [6]. Furthermore, alternative proteins address issues related to public health, food safety, and animal welfare [5, 6, 8, 9, 10]. However, there are challenges for their development. For instance, although cultivated meat uses significantly less land and water than conventional production, it may require higher energy inputs, depending on which specific chains are compared [11]. The climate effects of cultivated meat can also vary depending on the energy source used [8]. Biotechnological challenges for scaling up cultivated meat production in a cost-effective way remain [12], while the use of animal-free inputs presents its own set of research questions [13]. Yet another example is the need to understand consumer demands and intention to purchase cultivated meat products [14], as an evident essential part for the success of the new food production systems. Considering the remaining difficulties, the robust and multidimensional perceived benefits of cultivated meat and other alternative proteins boost increasing private and public investments for the resolution of the remaining challenges” (lines 66-84); “As a response to the need for science-based solutions, a new field of research, education, outreach activities and production systems conceived to develop and deliver alternative proteins, commonly denominated cellular agriculture, is currently under development [15]. In turn, alternative proteins are foods produced through the cultivation of animal cells ex vivo (i.e. out of the body), fermentation processes such as biomass and precision fermentation, or a combination of plant ingredients, with the goal of offering animal products or their analogues without the need for animal farming and slaughter [16]. Cultivated meat is a product of the culture of muscle tissue from a collection of cells from a live animal [17]” (lines 85-92); “This price-parity seems particularly difficult to achieve unless fundamental shifts in subsidies currently available for conventional meat are enacted [19]. It is known that price is a major driving force for consumer buying decisions [20]” (lines 98-100); “In addition, consumers who are skeptical when it comes to the decision of consuming cultivated meat report feelings of disgust, lack of naturalness, and negative sensory expectations [22, 23, 24, 25, 26, 27]. Food neophobia, defined as an individual's reluctance to consume new foods [18], is a general reason for rejecting any food item on the basis of its novelty and unfamiliarity which also plays a role for rejecting alternative proteins” (lines 106-111); “It is perceived that consumer acceptance will dictate its success [6, 30], in addition to being a determinant of the new balance between conventional and cultivated meat production chains. In this sense, whether cellular agriculture will become more than just a niche market [31] depends critically on the percentage of consumers that buy its products” (lines 119-123); “It is logical, then, that cultivated meat purchase intention be a relevant research topic. Indeed, the intention to consume cultivated meat has been the focus of scientific papers within the last decade, with first publications appearing in 2015 [32]. The levels of acceptance of cultivated meat vary among different research results worldwide [22, 33, 34, 35], as in Brazil [31, 36, 37, 38]” (lines 124-128).

Suggestion: The hypothesis about "significant differences" in the intention to consume cultivated meat across regions is somewhat vague. It would be helpful to specify potential influencing variables, such as food culture, economic status, or media exposure.

Response: To better explain the differences in the intention to consume cultivated meat across regions, we adapted the following paragraph: “Brazil hosts around half the South American population and around half its geographical area. In addition, Brazil is the largest exporter of beef and the second largest producer of meat in the world [39], being arguably considered a country with a barbecue culture [36]. As agriculture and animal production constitute a major component of the country’s Gross Domestic Product (GDP), the introduction of meat alternatives nationally, e.g. cultivated meat, seems essential to maintain its market share in the future, requiring careful planning to maximize the benefits and mitigate the disadvantages [40]. Thus, a more detailed regional understanding of Brazilians’ perceptions of and intention to consume alternative proteins, with their underlying reasons, seems both relevant and a rich study case scenario. Brazilian regions differ significantly in geographic, socioeconomic, food culture, and media exposure aspects and it is likely that such differences significantly influence consumer acceptance of new foods. Indeed, in a previous study comparing two Brazilian cities within the same geographical region of Brazil, the South, significant differences in consumption intention were observed [31], raising interesting scientific questions regarding the magnitude of differences which may appear when the comparison is expanded to include cities from other regions in the country” (lines 132-146) and “Salvador, the largest urban center in the Northeast region and the capital of the State of Bahia, with a population of 2,417,678 inhabitants [41], is currently ranked with the weakest economic and employment indicators among Brazilian State capitals [42]. In contrast, the city of São Paulo, capital of the Southeastern State of São Paulo, is the most populous city in Brazil, with a population of 11,451,99 inhabitants [41], and stands out as the most economically developed center in the country [42]” (lines 157-162).

Suggestion: To enhance the readability of the text, it would be beneficial to include an overview of the manuscript’s structure at the end of the Introduction Section. This would help guide readers through the content and provide a clearer understanding of how the study will unfold, making it easier to follow the main arguments and findings presented later in the paper.

Response: We summarized the structure of the manuscript at the end of the introduction: “This paper is structured to provide a demographic characterization of our participants, followed by descriptive and qualitative data obtained from their responses, which were later summarized and prepared for comparative and explanatory analyses, according to appropriate methodology for each case. Results were then statistically compared between cities and also analyzed considering relevant demographic data as potential explanatory factors. There follows a discussion section, organized from general to more specific issues, which includes the limitations of our study and potential priorities for further research in the field” (lines 163-170).

Suggestion: Material and Methods

The materials and methods section presents a clear and well-structured methodology, detailing the study's design, participant recruitment, and survey administration, which helps readers understand the research process. It emphasizes ethical considerations by including information about ethical approval and participant consent, adding credibility to the study. The comprehensive survey design, incorporating both multiple-choice and open-ended questions, allows for a nuanced understanding of consumer attitudes toward cultivated meat and facilitates the collection of both quantitative and qualitative data. Additionally, the robust statistical analysis methods used, such as binary logistic regression and ordinal regression, indicate a thorough approach to data analysis, enhancing the reliability of the findings. Although a list of questions is provided, a brief explanation of how specific questions relate to the study's objectives could help readers understand their significance.

Response: An explanation of how each questionnaire question relates to the study's objective was incorporated in: “In addition to questions specifically about cultivated meat acceptance, we selected related issues that may provide insights into respondents' acceptance levels. The degree of familiarity with cultivated meat is likely a key factor in understanding consumer interest. Similarly, capturing respondents' perceptions of the positive and negative impacts of conventional meat on environmental, human health, and animal welfare issues may help interpreting their motivations for accepting or rejecting meat alternatives. Thereby, questions 8 to 17 (Table 1) explore participants' level of knowledge about conventional and cultivated meat, their intention to consume cultivated meat, and their perceptions of how the production of either one affects the environment, human health, and animal welfare” (lines 186-195).

Suggestion: There is no discussion of potential biases in the sampling or response processes. Addressing limitations

---

## [Decision Letter · Decision Letter 1]

10 Dec 2024

PONE-D-24-43071R1How much do opinions regarding cultivated meat vary within the same country? The cases of Salvador and São Paulo, BrazilPLOS ONE

Dear Dr. Mendes,

Thank you for submitting your manuscript to PLOS ONE. After careful consideration, we feel that it has merit but does not fully meet PLOS ONE’s publication criteria as it currently stands. Therefore, we invite you to submit a revised version of the manuscript that addresses the points raised during the review process. **We invite you to follow the suggestions made by the two reviewers.**

Please submit your revised manuscript by Jan 24 2025 11:59PM. If you will need more time than this to complete your revisions, please reply to this message or contact the journal office at plosone@plos.org . Please include the following items when submitting your revised manuscript:

We look forward to receiving your revised manuscript.

Kind regards,

Giuseppina Rizzo

Academic Editor

PLOS ONE

**Journal Requirements:**

Reviewers' comments:

Reviewer's Responses to Questions

**Comments to the Author**

1. If the authors have adequately addressed your comments raised in a previous round of review and you feel that this manuscript is now acceptable for publication, you may indicate that here to bypass the “Comments to the Author” section, enter your conflict of interest statement in the “Confidential to Editor” section, and submit your "Accept" recommendation.

Reviewer #1: (No Response)

Reviewer #2: (No Response)

2. Is the manuscript technically sound, and do the data support the conclusions?

Reviewer #1: Yes

Reviewer #2: Yes

3. Has the statistical analysis been performed appropriately and rigorously? 

Reviewer #1: Yes

Reviewer #2: Yes

4. Have the authors made all data underlying the findings in their manuscript fully available?

Reviewer #1: Yes

Reviewer #2: Yes

5. Is the manuscript presented in an intelligible fashion and written in standard English?

Reviewer #1: Yes

Reviewer #2: Yes

6. Review Comments to the Author

**Reviewer #1: ** I would like to thank the Authors for following the suggestions provided in the first review. Despite their efforts, I believe that further revisions are needed to improve the overall quality of the manuscript. In particular:

1. In the introduction section, I appreciate the extensive references to the literature. However, I believe the introductory section should be kept distinct from the theoretical background section. That said, while the authors have included an overview of the paper's structure at the end of the introductory paragraph, it is not very clear. Therefore, I suggest that the authors rewrite this part to make it more concise and clearer.

2. The description of the methodology is clear in most parts, but some points can be enhanced for better understanding:

• The timing of the survey (September 5th to 23rd, 2019) is clearly stated, but it could be useful to briefly mention how the participants were selected within the cities, for example, whether they were chosen randomly, or if there were any stratifications or quotas used based on demographics.

• You mention considering various factors (city, income, education, and frequency of meat consumption) as explanatory variables. It would be beneficial to explain why these particular variables were selected. For example, what prior research or reasoning suggests that these variables are most likely to influence consumer attitudes toward cultivated meat?

• The statistical methods seem appropriate, but I would recommend providing a brief explanation or justification for the use of the "odds ratios" and "proportional odds regression models." For readers unfamiliar with these methods, a short explanation on why these specific models were chosen could be beneficial.

3. In my opinion, the revised version of the Discussion section is clearer and more concise. Despite that, while the key findings are highlighted, the connection between these findings and the broader context could be better clarified. In particular, explaining why respondents from Salvador and São Paulo differ in their knowledge and acceptance of cultivated meat could be strengthened by referencing social, economic, and cultural factors. For instance, you could explain that Salvador, with its higher unemployment rate and lower income levels, may have less access to information about new technologies, influencing their awareness and attitudes toward cultivated meat. Furthermore, I think that smoother transitions would improve the flow. For example, after discussing regional knowledge gaps, you could smoothly transition into the sensory properties of cultivated meat by noting how sensory factors, like taste and texture, play a critical role in consumer acceptance once familiarity with the product increases. You might use transitional phrases like "In addition to regional differences in awareness, sensory perceptions also shape acceptance," which would help link these ideas more cohesively.

**Reviewer #2:**  The authors have considered most of the suggestions. The changes incorporated significantly enhance the understanding of the study's contribution to the existing literature on this topic, highlight its relevance to the sector, and improve the presentation of the results through the incorporated tables. Detailed review is provided below:

• Suggestion: I recommend reconsidering the article's title in light of the question: Do opinions regarding cultivated meat vary within the same country? It seems self-evident that opinions differ among individuals, as these are shaped by a variety of factors. Even opinions within members of the same family can differ, so it is expected that opinions across a country, especially one as diverse as Brazil, would vary significantly due to its vast population and territory. Additionally, given Brazil's ex-tensive geographical area, putting “the case of Brazil” while the study to only two cities may not be advisable, as these cities may not be representative of the broader population. It may be useful to reflect this in the title, indicating that the study pertains to specific cities in Brazil- This suggestion was addressed.

The authors revised the title to 'How much do opinions regarding cultivated meat vary within the same country? The cases of Salvador and São Paulo, Brazil,' which clearly indicates that the study's conclusions are based on specific cities in Brazil and poses an open-ended question that is addressed through the study's findings.

• Suggestion: In the abstract and introduction, while the authors summarize the main research question and key findings, it would be beneficial to include more details about the research methodology used - This suggestion was not addressed.

I recommend including a concise description of the methods used to process and analyze data, as this information is currently missing from both abstract and introduction. For instance, quantitative methods such as binary logistic regression or ordinal regression models, and qualitative methods like the collective subject discourse methodology (CSD), could be briefly mentioned. The revised abstract is lengthy, I recommend balancing conciseness with essential information. In the introduction, the methods are only mentioned in the final paragraph as 'appropriate methodology for each case.' Adding a brief and clear description of the methods in this section would provide readers with a better understanding of the analytical approaches used in the study.

• Suggestion: Furthermore, while the authors mention that there is extensive literature on the subject, they do not clearly explain how this study relates to or builds upon previously published research- This suggestion was addressed.

The additional information enhances the study's context and underscores its importance within the broader body of literature on this topic.

• Regarding the results, I recommend presenting the results in tabular form for clarity and ease of interpretation- This suggestion was addressed.

The inclusion of tables significantly enhances the readability of the results. However, it would be beneficial to present each table either before or after a brief paragraph that summarizes the main findings, ensuring cohesion and flow within the text.

• Suggestion: For questions 10, 11, 15, and 17, if respondents have not previously heard about cultivated meat, may provide inconsistent responses due to their lack of knowledge on the subject, making the data less reliable for statistical analysis and reducing the comparability between respondents’ answers due to varied interpretations of the terms. Therefore, comparing responses from individuals familiar with cultivated meat to those from individuals unfamiliar with it may not be advisable. I suggest expanding the descriptive analysis to include segmentation, for example, those who consume meat versus those who do not, and those who are familiar with cultivated meat versus those who are not. Presenting a cross-tabulation that includes sociodemographic variables would also help to better understand the differences between these groups - This suggestion was partially addressed.

The paragraph included at the end of the discussion section effectively highlights the study's limitations, particularly those stemming from the challenges of researching public opinions on topics that are not widely known by the population.

The inclusion of Table 5, 'Responses to questions about the relationship between cultivated meat and the environment, human health, and animal welfare, based on respondents' familiarity with cultivated meat,' is valuable for identifying differences within the sample between respondents who answered YES and those who answered NO. However, this table requires a more in-depth interpretation or further mention in the discussion section to demonstrate its contribution to the study’s findings.

• Suggestion: Furthermore, it is important to provide detailed information regarding the steps taken to generate the word clouds, particularly the criteria used in the process. Care must be taken to avoid excessive reductionism, ensuring that the core meaning of the responses is retained. While using a thesaurus can be helpful in word cloud construction, its application must be handled carefully to avoid distorting the data. Additionally, it could be beneficial to conduct an additional analysis by excluding certain custom words and exploring new, potentially significant terms that may emerge in the word clouds - This suggestion was partially addressed.

The authors provide additional details about the methodology used to generate the word clouds. Since the suggestion to exclude certain custom words was not addressed, I recommend including a brief explanation to highlight the scope of the word cloud analysis.

7. PLOS authors have the option to publish the peer review history of their article (what does this mean? ). If published, this will include your full peer review and any attached files.

**Do you want your identity to be public for this peer review?** For information about this choice, including consent withdrawal, please see our Privacy Policy .

Reviewer #1: No

Reviewer #2: No

---

## [Author Response · Author response to Decision Letter 1]

18 Dec 2024

Dear Editor,

We sincerely appreciate your consideration of our manuscript and the constructive feedback provided by the reviewers. We are pleased to know that the work has merit, and we thank you for the invitation to submit a revised version addressing the points raised during the review process. We have carefully considered all recommendations and have addressed each one; we believe that the revisions resulted in a significantly improved manuscript.

We confirm that no changes were made to our financial disclosure statement. We inform you that the reference list has been reviewed to ensure it is complete and correct.

Dear Reviewers,

We sincerely thank you for your detailed evaluation and thoughtful feedback. Your valuable suggestions have greatly contributed to enhancing the quality and clarity of the manuscript. Each recommendation has been carefully reviewed and addressed, and we are confident that the revisions have substantially improved the manuscript.

Reviewer 1:

Suggestion: In the introduction section, I appreciate the extensive references to the literature. However, I believe the introductory section should be kept distinct from the theoretical background section. That said, while the authors have included an overview of the paper's structure at the end of the introductory paragraph, it is not very clear. Therefore, I suggest that the authors rewrite this part to make it more concise and clearer.

Comment: We have rewritten the overview of the paper's structure at the end of the introductory paragraph, making it more concise and clearer: “The paper begins by providing a demographic profile of the participants, followed by a detailed presentation of descriptive and qualitative data derived from their responses. These data are then summarized and analyzed using quantitative methods, including binary logistic regression and ordinal regression models, as well as the qualitative Collective Subject Discourse (CSD) [43] methodology. The analysis highlights comparisons between the two cities while considering demographic variables as potential explanatory factors. Finally, the discussion section addresses the study's findings, moving from general observations to more specific insights. It also outlines the study's limitations and suggests priorities for future research in the field” (lines 160-168).

Suggestion: The description of the methodology is clear in most parts, but some points can be enhanced for better understanding: The timing of the survey (September 5th to 23rd, 2019) is clearly stated, but it could be useful to briefly mention how the participants were selected within the cities, for example, whether they were chosen randomly, or if there were any stratifications or quotas used based on demographics.

Comment: We have now clarified that the participants were recruited randomly within the cities (line 173).

Suggestion: You mention considering various factors (city, income, education, and frequency of meat consumption) as explanatory variables. It would be beneficial to explain why these particular variables were selected. For example, what prior research or reasoning suggests that these variables are most likely to influence consumer attitudes toward cultivated meat?

Comment: We have expanded the discussion to explain the selection of these variables: “selected for their possible relationship with conventional meat consumption and the intention to consume cultivated meat, according to previous studies [13, 22, 29, 31]” (lines 185-187).

Suggestion: The statistical methods seem appropriate, but I would recommend providing a brief explanation or justification for the use of the "odds ratios" and "proportional odds regression models." For readers unfamiliar with these methods, a short explanation on why these specific models were chosen could be beneficial.

Comment: We have added a brief explanation to justify the use of odds ratios and proportional odds regression models. This addition clarifies that these methods were chosen for their ability to analyze ordinal outcomes and quantify the relationship between explanatory variables and the likelihood of different levels of consumer attitudes, making them particularly suitable for our study: “The statistical methods used, such as odds ratios and proportional odds regression models, are appropriate for analyzing the relationship between independent variables and the likelihood of categorical events, like the intention to consume cultivated meat. Odds ratios measure the effect of explanatory variables on event probabilities between two groups, while proportional odds regression models are useful for comparing outcomes with more than two ordered categories. These models were chosen for their ability to handle categorical data and proportional responses, common in consumption and attitude studies” (lines 225-231).

Suggestion: In my opinion, the revised version of the Discussion section is clearer and more concise. Despite that, while the key findings are highlighted, the connection between these findings and the broader context could be better clarified. In particular, explaining why respondents from Salvador and São Paulo differ in their knowledge and acceptance of cultivated meat could be strengthened by referencing social, economic, and cultural factors. For instance, you could explain that Salvador, with its higher unemployment rate and lower income levels, may have less access to information about new technologies, influencing their awareness and attitudes toward cultivated meat.

Comment: To improve the connection between the key findings and the broader context, we addressed the differences between respondents from Salvador and São Paulo, referring to social, economic, and cultural factors: “Salvador, with its higher unemployment rate and lower income levels, may have more limited access to information about new technologies, which in turn may influence the population's awareness and attitudes toward cultivated meat” (lines 359-361); “Residents of São Paulo live in a city with a more diversified economy and greater access to technologies and innovations, which likely offers more opportunities for being constantly informed about new food alternatives, such as cultivated meat. Additionally, the higher purchasing power and greater environmental awareness in São Paulo may positively influence the acceptance of alternative products, like cultivated meat. In contrast, citizens of Salvador, facing economic challenges such as a higher unemployment rate and lower purchasing power, may encounter barriers to accessing information about these new products, which could result in a lower intention to consume” (lines 374-382).

Suggestion: Furthermore, I think that smoother transitions would improve the flow. For example, after discussing regional knowledge gaps, you could smoothly transition into the sensory properties of cultivated meat by noting how sensory factors, like taste and texture, play a critical role in consumer acceptance once familiarity with the product increases. You might use transitional phrases like "In addition to regional differences in awareness, sensory perceptions also shape acceptance," which would help link these ideas more cohesively.

Comment: We revised the Discussion section to improve the flow by adding smoother transitions between ideas: “This trend is reflected in the intention to consume cultivated meat, which was lower” (line 365); “In addition to regional differences in awareness, sensory perceptions also influence the acceptance of cultivated meat. Apparently, people are motivated to try cultivated meat mainly out of curiosity regarding its organoleptic properties, such as taste and texture” (lines 384-386); “Moving from sensory perceptions to broader motivations, animal welfare concerns also play a significant role in the growing interest in consuming cultivated meat” (lines 397-398); “Alongside animal welfare, human health frequently drives the desire for cultivated meat. However, many respondents, particularly those unfamiliar with cultivated meat, believe conventional meat to be more health beneficial” (lines 409-411); “In terms of consumer demographics” (line 448); “While younger, wealthier, and more educated consumers tend to show more positive views towards cultivated meat, there are significant barriers to widespread acceptance, such as food neophobia” (lines 463-465); “As consumers become more familiar with and educated about cultivated meat, the transition from conventional to alternative protein sources may become smoother. This shift, however, is not without challenges” (lines 489-491).

Overall, we deeply appreciate the detailed evaluation, which was crucial in improving the work. We considered all the suggestions and made the necessary adjustments to strengthen the analysis and clarity of the results. All comments were carefully reviewed and incorporated to ensure the work met the high standards of excellence required by PLoS One.

Reviewer 2:

Suggestion: I recommend including a concise description of the methods used to process and analyze data, as this information is currently missing from both abstract and introduction. For instance, quantitative methods such as binary logistic regression or ordinal regression models, and qualitative methods like the collective subject discourse methodology (CSD), could be briefly mentioned. The revised abstract is lengthy, I recommend balancing conciseness with essential information.

Comment: We made the necessary adjustments, including a concise description of the methods used to process and analyze the data. In the abstract, we briefly mention the quantitative methods, such as binary logistic regression and ordinal regression models, as well as the qualitative method, the collective subject discourse methodology (CSD). We also adjusted the distribution of sections in the abstract, ensuring that all parts of the study were represented clearly and in balance.

“An online questionnaire comprising 17 multiple-choice and open-ended questions about opinions on conventional and cultivated meat was administered. The results were analyzed using quantitative methods, including binary logistic regression and ordinal regression models, as well as the qualitative Collective Subject Discourse methodology” (lines 31-34).

Suggestion: In the introduction, the methods are only mentioned in the final paragraph as 'appropriate methodology for each case.' Adding a brief and clear description of the methods in this section would provide readers with a better understanding of the analytical approaches used in the study.

Comment: We revised the introduction to include a brief and clear description of the methods used in the study: “The paper begins by providing a demographic profile of the participants, followed by a detailed presentation of descriptive and qualitative data derived from their responses. These data are then summarized and analyzed using quantitative methods, including binary logistic regression and ordinal regression models, as well as the qualitative Collective Subject Discourse (CSD) [43] methodology. The analysis highlights comparisons between the two cities while considering demographic variables as potential explanatory factors” (lines 160 -165).

Suggestion: The inclusion of tables significantly enhances the readability of the results. However, it would be beneficial to present each table either before or after a brief paragraph that summarizes the main findings, ensuring cohesion and flow within the text.

Comment: Thank you for the suggestion. We have implemented it in the first round of revisions by presenting the results in table format to provide greater clarity and facilitate interpretation. The key results from each table can be found in the paragraphs starting at lines 249, 260, 270, and 292, as well as in figures 1 and 2.

Suggestion: The paragraph included at the end of the discussion section effectively highlights the study's limitations, particularly those stemming from the challenges of researching public opinions on topics that are not widely known by the population.

The inclusion of Table 5, 'Responses to questions about the relationship between cultivated meat and the environment, human health, and animal welfare, based on respondents' familiarity with cultivated meat,' is valuable for identifying differences within the sample between respondents who answered YES and those who answered NO. However, this table requires a more in-depth interpretation or further mention in the discussion section to demonstrate its contribution to the study’s findings.

Comment: We made the necessary adjustments and included a more in-depth interpretation of Table 5 in the discussion section, highlighting its contribution to the study’s findings and the relevance of the differences between respondents who answered “yes” and those who answered “no”: “Whether or not they were familiar with cultivated meat, respondents generally believed that it is more beneficial than harmful to animal welfare” (lines 399-401); “People perceive that the introduction of cultivated meat to the market may reduce, or at least prevent the worsening of, animal suffering caused by factory farming, as this alternative meat production system does not require the slaughter of animals [22, 31]” (lines 404-407); “However, many respondents, particularly those unfamiliar with cultivated meat, believe conventional meat to be more health beneficial” (lines 410-411); “Prior knowledge of the similarities between cultivated and conventional meat, differing mainly in their production methods, may help clarify the perception that cultivated meat is more harmful to human health than conventional meat” (lines 421-424); “Even among those who were not familiar with cultivated meat, many considered it more beneficial than harmful to the environment” (lines 431-432); “as even those who were not yet familiar with it perceived it positively in terms of its environmental impact” (lines 442-443).

Suggestion: The authors provide additional details about the methodology used to generate the word clouds. Since the suggestion to exclude certain custom words was not addressed, I recommend including a brief explanation to highlight the scope of the word cloud analysis.

Comment: We included a brief explanation in the methodological section to highlight the scope of the word cloud analysis. Additionally, we clarified the criteria used in the selection of the words: “The selection of words for the cloud was based on two main criteria: their technical meaning and relevance in the context of the research, as well as their general meaning according to the dictionary” (lines 216-219); “For conventional meat, the main ideas in response to question 8 cited by the 809 respondents were i) production (21.6%), with words like farm, production, and reproduction; the word “production” was chosen because it encompasses the raising of animals for food, including reproduction and the process of breeding species, which is central to the meat industry; ii) suffering (11.7%), with citations such as cruelty, torture, suffering; the word "suffering" was chosen because it encompasses the terms mentioned by respondents when expressing their opinions on the treatment of animals in the conventional production system; iii) logistics (11.6%), which includes words such as distribution, transport, logistics; the word "logistics" reflects the process related to the logistical aspects of the conventional meat production supply chain. For cultivated meat, the main answers given involved i) innovation (12.1%), with words like technology, advance, innovative; the word "innovation" was highlighted due to its association with technological advances and the idea of something new and progressive mentioned by the respondents; ii) neophobia (10.8%), with words such as strange, fear, distrust; the word "neophobia" was chosen to represent the resistance and fear of new technologies expressed by the respondents; iii) artificial (8.2%), containing words such as false, unnatural, artificial; the word "artificial" was chosen as it is synonymous with the ideas of artificiality and unnaturalness expressed by the respondents” (lines 313-329).

Overall, we deeply a

---

## [Decision Letter · Decision Letter 2]

2 Jan 2025

PONE-D-24-43071R2How much do opinions regarding cultivated meat vary within the same country? The cases of São Paulo and Salvador, BrazilPLOS ONE

Dear Dr. Mendes, 

Thank you for submitting your manuscript to PLOS ONE. After careful consideration, we feel that it has merit but does not fully meet PLOS ONE’s publication criteria as it currently stands. Therefore, we invite you to submit a revised version of the manuscript that addresses the points raised during the review process.

Please submit your revised manuscript by Feb 16 2025 11:59PM. If you will need more time than this to complete your revisions, please reply to this message or contact the journal office at plosone@plos.org . Please include the following items when submitting your revised manuscript:

We look forward to receiving your revised manuscript.

Kind regards,

Giuseppina Rizzo

Academic Editor

PLOS ONE

Journal Requirements:

**Additional Editor Comments:**

We suggest you follow the advice of Reviewer 2 exactly in order to complete the referral process in the best possible way.

Reviewers' comments:

Reviewer's Responses to Questions

**Comments to the Author**

1. If the authors have adequately addressed your comments raised in a previous round of review and you feel that this manuscript is now acceptable for publication, you may indicate that here to bypass the “Comments to the Author” section, enter your conflict of interest statement in the “Confidential to Editor” section, and submit your "Accept" recommendation.

Reviewer #1: All comments have been addressed

Reviewer #2: (No Response)

2. Is the manuscript technically sound, and do the data support the conclusions?

Reviewer #1: Yes

Reviewer #2: Yes

3. Has the statistical analysis been performed appropriately and rigorously? 

Reviewer #1: Yes

Reviewer #2: Yes

4. Have the authors made all data underlying the findings in their manuscript fully available?

Reviewer #1: Yes

Reviewer #2: Yes

5. Is the manuscript presented in an intelligible fashion and written in standard English?

Reviewer #1: Yes

Reviewer #2: Yes

6. Review Comments to the Author

Reviewer #1: The authors have thoroughly addressed all comments and suggestions provided during the review process.

In my opinion, there is no further need for improvement.

As the paper also meets the publication criteria of PLOS ONE, I consider it accepted for publication.

Reviewer #2: The authors have addressed most of the suggestions. The changes significantly enhance the understanding of methodological aspects and enrich the discussion of the findings. Nonetheless, it is essential to incorporate a more comprehensive analysis of the results and enhance their articulation. These improvements will require corresponding revisions in the discussion section. A detailed review is provided below:

Suggestion: I recommend including a concise description of the methods used to process and analyze data, as this information is currently missing from both abstract and introduction. For instance, quantitative methods such as binary logistic regression or ordinal regression models, and qualitative methods like the collective subject discourse methodology (CSD), could be briefly mentioned. The revised abstract is lengthy, I recommend balancing conciseness with essential information.

This suggestion has been addressed. The abstract now includes a brief description of the analytical approaches used in the study.

Suggestion: In the introduction, the methods are only mentioned in the final paragraph as 'appropriate methodology for each case.' Adding a brief and clear description of the methods in this section would provide readers with a better understanding of the analytical approaches used in the study.

This suggestion has been addressed. The introduction now includes a brief description of the analytical approaches used in the study.

Suggestion: The inclusion of tables significantly enhances the readability of the results. However, it would be beneficial to present each table either before or after a brief paragraph that summarizes the main findings, ensuring cohesion and flow within the text.

This suggestion remains unaddressed. Although the authors have added tables in the initial revisions, the latest suggestion to enhance the description of the results presented in these tables has not been considered. I recommend identifying key insights (including those related to demographics) and comparing data between cities, given the paper's emphasis on the variability of opinions within the same country. This analysis is critical and needs careful articulation to ensure clarity, smooth transitions, and emphasis on key findings. Additionally, the titles of tables and figures contain unnecessary details such as “809 respondents in an online questionnaire from September 5 to 23, 2019” . Such specifics should be presented in the materials and methods section.

Suggestion: The inclusion of Table 5, 'Responses to questions about the relationship between cultivated meat and the environment, human health, and animal welfare, based on respondents' familiarity with cultivated meat,' is valuable for identifying differences within the sample between respondents who answered YES and those who answered NO. However, this table requires a more in-depth interpretation or further mention in the discussion section to demonstrate its contribution to the study’s findings.

The authors have addressed this suggestion by identifying relevant differences and similarities between both groups in the discussion section. However, as this point relates to the previous suggestion, I recommend first clarifying these key insights in the results section. They should then be further analyzed within the discussion section to enhance the understanding of their implications in the broader context.

Suggestion: The authors provide additional details about the methodology used to generate the word clouds. Since the suggestion to exclude certain custom words was not addressed, I recommend including a brief explanation to highlight the scope of the word cloud analysis.

The authors have provided valuable clarifications about the technical criteria, which enhance the understanding of the process used to generate the word clouds. However, the contribution of these word clouds to the overall study remains unclear, as there is no analysis of them in the results section. Although their primary function—to identify the most frequent words—might seem obvious, explicitly stating and discussing this purpose within the results would clarify their role in the study.

7. PLOS authors have the option to publish the peer review history of their article (what does this mean? ). If published, this will include your full peer review and any attached files.

**Do you want your identity to be public for this peer review?** For information about this choice, including consent withdrawal, please see our Privacy Policy .

Reviewer #1: No

Reviewer #2: No

---

## [Author Response · Author response to Decision Letter 2]

3 Jan 2025

Dear Editor,

We would like to express our sincere gratitude for your consideration of our manuscript and the constructive feedback provided by the reviewers. We are pleased that the work has been well-received, and we appreciate the invitation to submit a revised version that addresses the points raised during the review process. We have thoroughly reviewed all the recommendations and made the necessary revisions, which we believe have substantially improved the manuscript.

We confirm that no changes have been made to our financial disclosure statement. Additionally, we have reviewed the reference list to ensure its accuracy and completeness.

Dear Reviewers,

We deeply appreciate your thorough evaluation and insightful feedback. Your invaluable suggestions have significantly contributed to improving both the quality and clarity of the manuscript. Each of your recommendations has been carefully considered and incorporated, and we are confident that the revisions have greatly enhanced the manuscript.

Reviewer 1:

Suggestion: The authors have thoroughly addressed all comments and suggestions provided during the review process. In my opinion, there is no further need for improvement. As the paper also meets the publication criteria of PLOS ONE, I consider it accepted for publication.

Response: Dear reviewer 1, we would like to express our sincere and profound gratitude for the valuable suggestions and comments provided. Your detailed and relevant observations undoubtedly played a key role in the substantial improvement of the work, allowing us to enhance several important aspects of the research. We greatly appreciate the time and effort dedicated to reviewing the manuscript, and we are confident that the modifications made in response to your recommendations have significantly strengthened the quality and clarity of the content presented.

Reviewer 2:

Suggestion: Although the authors have added tables in the initial revisions, the latest suggestion to enhance the description of the results presented in these tables has not been considered. I recommend identifying key insights (including those related to demographics) and comparing data between cities, given the paper's emphasis on the variability of opinions within the same country. This analysis is critical and needs careful articulation to ensure clarity, smooth transitions, and emphasis on key findings. Additionally, the titles of tables and figures contain unnecessary details such as “809 respondents in an online questionnaire from September 5 to 23, 2019”. Such specifics should be presented in the materials and methods section.

Response: We adapted the demographic data table (Table 2) to compare information between cities. Additionally, we included a brief paragraph before each table, highlighting the main findings: “When respondents were asked if they had heard about cultivated meat, 310 out of 809 participants (38.3%) answered yes (Fig. 1). In São Paulo, 187 respondents (44.6%) reported familiarity with cultivated meat, compared to 123 (31.5%) in Salvador. Table 3 presents the results of this question broken down by demographic variables” (lines 258-261); “Overall, 265 participants (32.8%) expressed interest in eating cultivated meat, 211 (26.0%) indicated they would not consume it, and 333 (41.2%) responded "I don't know" (Fig. 1). In São Paulo, 170 respondents (40.6%) said they would eat cultivated meat, compared to 95 respondents (24.4%) in Salvador. Table 4 presents the results of this question, categorized by demographic variables” (lines 268-272); “Among respondents who were already familiar with cultivated meat, 205 (66.1%) believe it benefits animal welfare, 108 (34.8%) consider it beneficial for human health, and 174 (56.1%) for the environment. In contrast, 215 (43.1%) among those who had never heard of cultivated meat, believe it benefits animal welfare, 75 (15.0%) see benefits for human health, and 199 (39.9%) for the environment (Table 5)” (lines 294-298).

As for the details provided in the title of tables, we agree that this information has already been presented in the Material and Methods section. However, we decided to maintain it because we understand that it is important that readers get the full information when looking at tables or figures, independently from reading the text. We also want to consider the opinion of the reviewer who has already accepted the paper as it is, by avoiding the deletion of any part of the manuscript.

Suggestion: The authors have addressed this suggestion by identifying relevant differences and similarities between both groups in the discussion section. However, as this point relates to the previous suggestion, I recommend first clarifying these key insights in the results section. They should then be further analyzed within the discussion section to enhance the understanding of their implications in the broader context.

Response: We clarified the main findings from Table 5 in the Results section, structuring them according to the sequence in which they are discussed throughout the text. In the Discussion section, we address the results related to animal welfare, human health, and the environment, between lines 406 and 454.

Suggestion: The authors have provided valuable clarifications about the technical criteria, which enhance the understanding of the process used to generate the word clouds. However, the contribution of these word clouds to the overall study remains unclear, as there is no analysis of them in the results section. Although their primary function—to identify the most frequent words—might seem obvious, explicitly stating and discussing this purpose within the results would clarify their role in the study.

Response: We explained the purpose and role of Figure 3, which presents the word clouds, in the Results section, highlighting its contribution to the data analysis: “The respondents were asked to share the three words that came to their minds when thinking about the production systems of conventional and cultivated meat. Given the large volume of data, after analyzing each term, a word cloud was generated with the words that best represented the ideas expressed by the participants. For conventional meat, the words that best reflected the ideas mentioned by the participants were…” (lines 316-320); “For cultivated meat, the words that best reflected the ideas mentioned by the participants were…” (lines 328-329).

In summary, we sincerely appreciate the meticulous evaluation and insightful suggestions, which have been instrumental in enhancing this work. Each comment and recommendation was carefully reviewed, and the revisions implemented are intended not only to address the observations but also to strengthen the clarity and depth of the research. We are grateful for the time and effort invested in the review process, as well as for the constructive and enriching feedback provided.

Please do not hesitate to contact us if any further information is required.

Sincerely,

The Authors.

---

## [Editor Report · Decision Letter 3]

7 Jan 2025

How much do opinions regarding cultivated meat vary within the same country? The cases of São Paulo and Salvador, Brazil

PONE-D-24-43071R3

Dear Dr. Mendes,

We’re pleased to inform you that your manuscript has been judged scientifically suitable for publication and will be formally accepted for publication once it meets all outstanding technical requirements.

Kind regards,

Giuseppina Rizzo

Academic Editor

PLOS ONE
---

## [Editor Report · Acceptance letter]

PONE-D-24-43071R3

PLOS ONE

Dear Dr. Mendes,

I'm pleased to inform you that your manuscript has been deemed suitable for publication in PLOS ONE. Congratulations! Your manuscript is now being handed over to our production team.

Kind regards,

on behalf of

Dr. Giuseppina Rizzo

Academic Editor

PLOS ONE